# Dose-Dependent Effects of Radiation on Mitochondrial Morphology and Clonogenic Cell Survival in Human Microvascular Endothelial Cells

**DOI:** 10.3390/cells13010039

**Published:** 2023-12-23

**Authors:** Li Wang, Rafael Rivas, Angelo Wilson, Yu Min Park, Shannon Walls, Tianzheng Yu, Alexandra C. Miller

**Affiliations:** 1Armed Forces Radiobiology Research Institute, Uniformed Services University of the Health Sciences, Bethesda, MD 20889, USA; li.wang.ctr1@usuhs.edu (L.W.); rafael.rivas@usuhs.edu (R.R.); angelo.wilson.ctr@usuhs.edu (A.W.); walls.shannon.d@gmail.com (S.W.); 2Department of Pathology, Uniformed Services University of the Health Sciences, Bethesda, MD 20889, USA; 3Henry M Jackson Foundation for the Advancement of Military Medicine, Bethesda, MD 20817, USA; yumin.park@wsu.edu (Y.M.P.); tianzheng.yu.ctr@usuhs.edu (T.Y.); 4Consortium for Health and Military Performance, Department of Military and Emergency Medicine, Uniformed Services University of the Health Sciences, Bethesda, MD 20814, USA; 5Department of Radiation Science and Radiology, Uniformed Services University Health Sciences, Bethesda, MD 20889, USA; 6Columbia University Irving Medical Center, Columbia University, New York, NY 10032, USA

**Keywords:** radiation exposure, dose-dependent effect, human microvascular endothelial cells, cell survival, mitochondrial morphology and function, DNA damage, reactive oxygen species, cell senescence

## Abstract

To better understand radiation-induced organ dysfunction at both high and low doses, it is critical to understand how endothelial cells (ECs) respond to radiation. The impact of irradiation (IR) on ECs varies depending on the dose administered. High doses can directly damage ECs, leading to EC impairment. In contrast, the effects of low doses on ECs are subtle but more complex. Low doses in this study refer to radiation exposure levels that are below those that cause immediate and necrotic damage. Mitochondria are the primary cellular components affected by IR, and this study explored their role in determining the effect of radiation on microvascular endothelial cells. Human dermal microvascular ECs (HMEC-1) were exposed to varying IR doses ranging from 0.1 Gy to 8 Gy (~0.4 Gy/min) in the AFRRI 60-Cobalt facility. Results indicated that high doses led to a dose-dependent reduction in cell survival, which can be attributed to factors such as DNA damage, oxidative stress, cell senescence, and mitochondrial dysfunction. However, low doses induced a small but significant increase in cell survival, and this was achieved without detectable DNA damage, oxidative stress, cell senescence, or mitochondrial dysfunction in HMEC-1. Moreover, the mitochondrial morphology was assessed, revealing that all doses increased the percentage of elongated mitochondria, with low doses (0.25 Gy and 0.5 Gy) having a greater effect than high doses. However, only high doses caused an increase in mitochondrial fragmentation/swelling. The study further revealed that low doses induced mitochondrial elongation, likely via an increase in mitochondrial fusion protein 1 (Mfn1), while high doses caused mitochondrial fragmentation via a decrease in optic atrophy protein 1 (Opa1). In conclusion, the study suggests, for the first time, that changes in mitochondrial morphology are likely involved in the mechanism for the radiation dose-dependent effect on the survival of microvascular endothelial cells. This research, by delineating the specific mechanisms through which radiation affects endothelial cells, offers invaluable insights into the potential impact of radiation exposure on cardiovascular health.

## 1. Introduction

Microvascular endothelial cells (ECs) can be significantly impacted by exposure to radiation [1,2], with the effect varying depending on the dose exposed [3,4,5]. High doses of radiation can directly damage ECs, resulting in the increased permeability of blood vessels [6,7,8], impaired blood flow [2,3], reduced tissue oxygenation [9], and the release of proinflammatory cytokines and chemokines [10,11]. This leads to radiation-induced tissue damage, including fibrosis [12,13,14] and necrosis [15,16], which can cause negative health effects such as cardiovascular diseases [17,18], neurological disorders [19], and cancer [20]. When using radiation therapy to treat cancer, the effects of radiation on microvascular ECs in normal tissues are particularly concerning due to the potential adverse effects, including impaired wound healing [21], chronic inflammation [22], and tissue fibrosis [23,24]. Compared to the well-documented high-dose effects, the effects of low-dose radiation (≤0.1 Gy) on ECs are subtle but more complex, and still under debate. Low-dose (<1 Gy) radiation mentioned here refers to radiation exposure levels that are below those that cause immediate and noticeable damage. The potential negative consequences of low-dose radiation on endothelial cells include impairing their ability to regulate vasodilation and vasoconstriction [25,26] and compromising the blood–tissue barrier function [27]. However, some studies have suggested that low doses of radiation may trigger repair mechanisms in microvascular ECs to counteract the damage and restore normal functions. In limited studies, low-dose radiation exposure has been shown to have a beneficial effect on ECs, such as anti-inflammatory [28,29] and pro-angiogenic [30] effects, which is known as radiation hormesis [31,32,33]. However, the potential beneficial effects of low-dose radiation are still debated and additional research is needed to fully understand the role that low-dose radiation plays in the development of radiation-induced effects.

Mitochondria are known for generating adenosine triphosphate (ATP). However, they also play a role in the development of endothelial dysfunction [34]. Damaged mitochondria in the endothelium can mediate endothelial dysfunction via three possible mechanisms, including the overproduction of reactive oxygen species (ROS) [34], the reduced production of nitric oxide (NO) [35], and impaired energy production [36]. The overproduction of ROS can result in inflammation and oxidative stress, leading to endothelial damage and dysfunction [37]. NO, on the other hand, is an essential signaling molecule that promotes vasodilation and regulates blood vessel function. A reduction in NO production can further impair endothelial and blood vessel function [37]. In addition, the decreased availability of ATP due to impaired energy production can also contribute to endothelial dysfunction [37]. Endothelial dysfunction can exacerbate mitochondrial dysfunction by increasing oxidative stress [38]. Therefore, the relationship between mitochondrial dysfunction and endothelial dysfunction is complicated.

Recent studies have indicated that exposure to radiation causes significant damage to mitochondria [39,40]. The damage to mitochondria can be brought about by multiple mechanisms, such as mitochondrial DNA lesions [39], changes in mitochondrial membrane composition [41], and the modification of mitochondrial protein expression [42,43]. Radiation and ROS can cause more damage to mitochondrial DNA compared to nuclear DNA because mitochondrial DNA lacks histone protection [39,44,45,46,47]. Additionally, the mitochondrial genome does not possess an efficient DNA repair mechanism, making it more prone to accumulating unrepaired lesions than its nuclear counterpart [39,48,49,50]. The opening of the mitochondrial permeability transition pore (MPTP) due to radiation-induced damage is another consequence that leads to mitochondrial swelling, a reduction in mitochondrial membrane potential, and the release of pro-apoptotic proteins like cytochrome c, resulting in mitochondrial permeability transition (MPT) [41]. Moreover, studies indicate that radiation exposure can alter the expression of mitochondrial proteins involved in the electron transport chain and oxidative phosphorylation [42,43], ultimately leading to mitochondrial dysfunction.

The impact of radiation on mitochondria and the endothelium is well established, and there is a close association between mitochondrial damage and endothelial dysfunction. This led us to hypothesize that changes in mitochondria may underlie the dose-dependent endothelial response triggered by radiation exposure. As a result, this research aimed to explore the dose-dependent consequences of radiation exposure on microvascular endothelial cells, which are crucial components of blood vessels and play a vital role in cardiovascular health [51]. Additionally, our investigation delved into the role of the mitochondrial morphology and function in determining the effect of radiation on microvascular endothelial cells.

## 2. Materials and Methods

### 2.1. Primary Antibodies

The following commercial antibodies were used in the Western blot experiments: anti-p21 (WAF1, Cip1) (#14-6715-81, Invitrogen, Carlsbad, CA, USA), anti-p16INK4α (#PA5-20379, Invitrogen, Carlsbad, CA, USA), anti-p53 (#ab131442, abcam, Cambridge, MA, USA), anti-β actin (#A5441, Sigma-Aldrich, St. Louis, MO, USA), anti-Mfn1 (#14739, Cell Signaling Technology, Danvers, MA, USA), anti-Mfn2 (#9482, Cell Signaling Technology, Danvers, MA, USA), anti-Opa1 (#612606, BD Biosciences, San Jose, CA, USA), anti-VDAC (#4661, Cell Signaling Technology, Danvers, MA, USA), anti-Drp1 (#611112, BD Biosciences, San Jose, CA, USA), anti-GAPDH (#ABS16, Sigma-Aldrich, St. Louis, MO, USA), Ser139_phospho detection antibody H2AX (#PEL-H2AX-S139-T, RayBiotech, Norcross, GA, USA), pan detection antibody H2AX (#PEL-H2AX-S139-T, RayBiotech, Norcross, GA, USA).

### 2.2. Cell Culture

In this experiment, a male patient-derived human dermal microvascular endothelium (HMEC-1) cell line was utilized. The HMEC-1 cells (#CRL-3243) were procured from the American Type Culture Collection (ATCC, Manassas, VA, USA) and were maintained in MCDB131 (#10372019, ThermoFisher Scientific, Waltham, MA, USA) complete growth medium supplemented with 10% fetal bovine serum (FBS; #30-2020, ATCC, Manassas, VA, USA), 10 ng/mL epidermal growth factor (EGF; #PHG0314, ThermoFisher Scientific, Waltham, MA, USA), 1 µg/mL hydrocortisone (#H0396, Sigma, St. Louis, MO, USA), 10 mM glutamine (#302214, ATCC, Manassas, VA, USA), and 100 units/mL of penicillin and 100 µg/mL of streptomycin (#15140122, ThermoFisher Scientific, Waltham, MA, USA) at 37 °C under a humid atmosphere of 5% CO_2_/95% air, as per the manufacturer’s instructions. The complete growth medium was refreshed twice per week. 

For irradiation (IR) experiments, HMEC-1 cells (at passage 7) were seeded in T25 flasks at a density of approximately 3 × 10^6^ cells per flask and cultured for three to four days prior to IR. The day before IR, when the cultures were in the early plateau phase, the complete growth medium was substituted with fresh cell starvation medium, which contained MCDB131, 0.1% FBS, 1 µg/mL hydrocortisone, 10 mM glutamine, and 100 units/mL of penicillin and 100 µg/mL of streptomycin.

### 2.3. Irradiation

The HMEC-1 cultures in the plateau phase were subjected to gamma radiation at doses ranging from 0.1 Gy to 8 Gy, using a dose rate of 0.4 Gy/min, at the Armed Forces Radiobiology Research Institute (AFRRI) 60-Cobalt facility, as previously described [52]. Following radiation, the cells were detached using Accutase^®^ cell detachment solution (#25-058-CI, Corning, Corning, NY, USA), counted using the CountessTM II Automated Cell Counter (Catalog #, AMQAX1000, Invitrogen, Carlsbad, CA, USA), and re-plated in complete growth medium if the sub-culturing of cells in 6-well plates or 10-cm dishes was necessary. It should be noted that, to minimize DNA damage repair post-IR, the irradiated or sham-irradiated T25 flasks were temporarily placed on ice before cell harvest if they could not be handled immediately.

### 2.4. Clonogenic Growth Assay

The standard clonogenic growth assay was used to measure the survival of irradiated HMEC-1 cells [53]. After IR, the cells were harvested, suspended in MCDB131 complete growth media, and then diluted and re-plated in triplicate in 100-mm cell culture dishes. The goal was to form 30–100 surviving colonies over a 14-day culture period without a media change. After 14 days, the colonies were fixed with 100% methanol for 20 min at room temperature and stained with a 1:20 dilution of Giemsa Stain solution (#G5637, Sigma-Aldrich, Saint Louis, MO, USA) according to the manufacturer’s instructions. A colony was defined as a cell cluster consisting of at least 60 cells. The plating efficiency (PE) was calculated by dividing the number of colonies formed for non-irradiated and irradiated cells by the number of cells seeded and then multiplying by 100%. The surviving fraction (SF) was normalized from the PE for irradiated cells to the PE for non-irradiated cells. The SF was measured in N = 3–32 samples, and data analysis was conducted using one-way ANOVA and Dunnett’s multiple comparisons test. A significance level of * *p* < 0.05 was used.

### 2.5. Protein Sample Preparation and Western Blot Analysis

The HMEC-1 cells were collected and homogenized using ice-cold Pierce^TM^ Radio-Immunoprecipitation Assay (RIPA) lysis and extraction buffer (#89901, ThermoFisher Scientific, Waltham, MA, USA), supplemented with the Halt^TM^ protease inhibitor (Prod #1860932, ThermoFisher Scientific, Waltham, MA, USA) or Halt^TM^ protease and phosphatase inhibitor cocktail (Prod #1861281, ThermoFisher Scientific, Waltham, MA, USA), via the freeze–thaw lysis method combined with sonication. The cell samples were incubated at 4 °C for 30–60 min with gentle rotation before centrifugation at 14,000 rpm at 4 °C for 15–30 min. The protein concentration was measured using the bicinchoninic acid (BCA) protein assay kit (#7780; Cell Signaling Technology, Danvers, MA, USA) according to the manufacturer’s instructions. Total cell lysates (10–30 µg) from the irradiated or sham-irradiated HMEC-1 cells were separated by Novex^®^ NuPAGE SDS-PAGE (ThermoFisher Scientific, Waltham, MA, USA) and transferred to nitrocellulose membranes (ThermoFisher Scientific, Waltham, MA, USA). The transferred membranes were washed with 1× Tris-buffered saline with 0.1% Tween^®^ 20 detergent (1× TBST), blocked with 5% milk/1× TBST, and probed with primary antibodies (diluted in 3% BSA/1× TBST) followed by appropriate secondary antibodies conjugated with horseradish peroxidase (HRP) (Goat-anti-Mouse IgG HRP-conjugated, #HAF007; Goat-anti-Rabbit IgG HRP-conjugated, #HAF008; R&D Systems, Minneapolis, MN, USA) diluted in 5% milk/1× TBST. The Pierce^TM^ ECL Western blotting substrate (Catalog #32209, ThermoFisher Scientific, Waltham, MA, USA) or Pierce^TM^ ECL plus Western blotting substrate (Catalog #32132, ThermoFisher Scientific, Waltham, MA, USA) was used for signal detection. Densitometry evaluation was performed using Image Lab^TM^ (Version 6.0.1, Standard Edition, Bio-Rad Laboratories, Inc., Philadelphia, PA, USA). For P53, the sample size was N = 2 and data analysis was conducted using the Kruskal–Wallis test followed by Dunn’s multiple comparisons test, where * *p* < 0.05. For others, the sample size was N = 3–4, and data analysis was conducted using one-way ANOVA and Dunnett’s multiple comparisons test, where * *p* < 0.05.

### 2.6. Phospho-H2AX (Ser139) and Total H2AX 

The HMEC-1 cells were irradiated with different doses of IR (0 Gy, 0.25 Gy, and 8 Gy) and then lysed to obtain cell lysates. To measure the levels of phospho-H2AX and total H2AX in the lysates at 1 h after IR, ELISA was performed using the RayBio^®^ phospho-H2AX (Ser139) and total H2AX ELISA kit (#PEL-H2AX-S139-T, RayBiotech, Norcross, GA, USA), following the manufacturer’s instructions. Western blot analysis was also performed using the same detection antibodies provided in the ELISA kit to confirm the ELISA results. To standardize the results for both ELISA and Western blot analysis, the absorbance values at 450 nm for phospho-H2AX in ELISA and the band intensity for phospho-H2AX in Western blot analysis were adjusted by dividing them with the optical density values and band densimetry values for pan-H2AX, respectively. The resulting values were expressed as rH2AX/H2AX for each sample. The sample size was N = 4 per radiation dose. The results were analyzed using one-way ANOVA and Dunnett’s multiple comparisons test, with a significance level of * *p* < 0.05. 

### 2.7. Senescence β-Galactosidase Activity Assay

The Cell Signaling Technology senescence β-galactosidase activity assay kit (Fluorescence, plate-based, # 23833, Danvers, MA) was utilized to measure senescence-associated β-galactosidase (SA-β-gal) activity at pH 6.0, which is a commonly recognized marker of replicative senescence. The assay was performed as per the manufacturer’s protocol. Briefly, HMEC-1 cells that were harvested four days after IR were lysed in cold 1× senescence cell lysis buffer containing 4 mM Pefabloc (#1142986800, Sigma, St. Louis, MO, USA) and 1× Halt^TM^ protease/phosphatase inhibitor (#1861281, ThermoFisher Scientific, Waltham, MA). The lysate was then prepared by scraping the cells off the surface and pipetting the lysate up and down a few times. The cell lysate was obtained through centrifugation at 14,000 rpm for 15 min at 4 °C. The total protein concentration was determined using the BCA assay mentioned previously. The cell lysate, with a protein concentration of 0.25 mg/mL (50 µL), was mixed with 2× senescence reaction buffer containing 10 mM β-mercaptoethanol and 1× SA-β-gal substrate (50 µL) in a 96-well plate and then incubated at 37 °C in the dark. During the 3-h incubation, the substrate, 4-methylumbelliferyl β-D-galactopyranoside (4-MUG), was hydrolyzed by β-gal in the cell lysate to 4-methylumbelliferone (4-MU), which was measured at λ_ex_/λ_em_ = 360/465 nm using a fluorescent plate reader (CLARIOstar microplate reader, BMG Labtech, Cary, NC, USA). The SA-β-gal activity in the total cell lysate correlated with the fluorescent intensity. The assay was performed in duplicate, with N = 3–6 per radiation dose. Data analysis was performed using one-way ANOVA and Dunnett’s multiple comparisons test, with significance set at * *p* < 0.05.

### 2.8. Measurements of Cellular Levels of Reactive Oxygen Species (ROS)

The Invitrogen^TM^ dichlorodihydrofluorescein diacetate (DCFDA, #C6827, ThermoFisher Scientific, Waltham, MA, USA) was used to directly detect ROS levels in live cells [54], following the manufacturer’s instructions. DCFDA is chemically reduced and can be converted to the fluorescent 2′,7′-dichlorofluorescein (DCF) by intracellular esterase cleavage and oxidation. The cultures were incubated with DCFDA (5 µM) for 30 min at 37 °C, and the fluorescent green intensity, which represented oxidized DCF intercalated within the cell’s DNA, was used to quantify the cellular ROS levels. Fluorescence images were captured using a Nikon Eclipse Ti epifluorescence microscope with a digital camera (Melville, NY, USA) at λ_ex_/λ_em_ = 495/525 nm, and the fluorescent intensity was measured using the NIH ImageJ software. Images underwent adjustments for sharpness and contrast, and these modifications were uniformly applied to the entire image. The study included N = 3 for all doses. Data analysis was performed using one-way ANOVA and Dunnett’s multiple comparisons test, with statistical significance set at * *p* < 0.05. 

### 2.9. Assessment of Mitochondrial Membrane Potential (ΔΨ_m_)

The assessment of mitochondrial membrane potential (ΔΨm) in live cells was conducted using the cell-permeant cationic fluorescent dye tetramethylrhodamine ethyl ester (TMRE), obtained from ThermoFisher Scientific (#T669, Waltham, MA, USA) [55]. The manufacturer’s instructions were followed, and TMRE was added directly to the cell culture medium at a concentration of 100 nM. After incubation for 15 min at 37 °C under light-protected conditions, the red–orange fluorescent dye’s intensity sequestered by active mitochondria was used to determine ΔΨm. Fluorescence images were captured using a Nikon Eclipse Ti epifluorescence microscope with a digital camera (Melville, NY, USA) at λ_ex_/λ_em_ = 555/613 nm. The fluorescent intensity was analyzed with the ImageJ software (NIH). Images underwent adjustments for sharpness and contrast, and these modifications were uniformly applied to the entire image. The study included N = 3 for all doses. The data were analyzed using one-way ANOVA and Dunnett’s multiple comparisons test. The significance level was set at * *p* < 0.05.

### 2.10. Assessment of Mitochondrial Morphology 

To visualize mitochondria, a cell-permeable probe called MitoTracker Red chloromethyl-X-rosamine (CMXRos), obtained from ThermoFisher Scientific (#M7512, Waltham, MA, USA), was utilized, following the manufacturer’s instructions. MitoTracker Red CMXRos (100 nM) was added directly to the cell culture medium and incubated for 15 min at 37 °C under light-protected conditions. Fluorescence images were captured using a Nikon Eclipse Ti epifluorescence microscope with a digital camera (Melville, NY) at λ_ex_/λ_em_ = 555/613 nm. Images underwent adjustments for sharpness and contrast, and these modifications were uniformly applied to the entire image. The study included N = 3 for all doses. The data were analyzed using one-way ANOVA and Dunnett’s multiple comparisons test, and the significance level was set at * *p* < 0.05.

### 2.11. Statistical Analysis

The data are expressed as the mean ± standard error of the mean (SEM). Statistical analyses were performed using the GraphPad Prism 10.1.2 (324) software, and mean group differences were compared using a one-way ANOVA with Dunnett’s multiple comparisons test or the Kruskal–Wallis test followed by Dunn’s multiple comparisons test. A significance level of *p* < 0.05 was used to determine statistical significance.

## 3. Results

### 3.1. Radiation Exposure Affects Cell Survival in Irradiated HMEC-1

The growth of endothelial cells is affected by radiation exposure, particularly their ability to divide and form new colonies. This is crucial in promoting angiogenesis [56,57] and vascular repair [58] and maintaining cardiovascular health [59]. While the effects of radiation on macrovascular endothelial cells are well documented [60,61], less is known about how microvascular endothelial cells are affected by radiation. In this study, HMEC-1 cells were cultured in low-serum media and exposed to various doses of Cobalt-60 gamma radiation. The survival of these cells was assessed following a 14-day period using a clonogenic growth assay. The results showed that radiation exposure significantly reduced survival in HMEC-1 cells at high doses (equal to or higher than 1 Gy), as demonstrated by an exponential decrease in colony numbers (Figure 1A). However, low doses of radiation (less than 1 Gy), specifically 0.25 Gy, resulted in a 15% increase in colony number compared to control cells (Figure 1B). It should be noted that while both 0.1 Gy and 0.25 Gy resulted in similar increases in colony number, the results with 0.1 Gy were more variable and not statistically significant (Figure 1B). Additionally, there was no difference in colony number between cells exposed to 0.5 Gy of radiation and the non-irradiated cells (Figure 1B).

### 3.2. High Doses Increase γH2AX and P53 in the Irradiated HMEC-1

In cells, the phosphorylation of H2AX at its serine 139 residue (rH2AX) and the activation of p53 are crucial responses to DNA damage caused by radiation, particularly double-strand breaks [62,63,64]. These responses aid in the repair of damaged DNA or remove cells with significant damage through apoptosis [62,63,64]. To investigate this phenomenon, we assessed rH2AX/H2AX in HMEC-1 exposed to 0 Gy, 0.25 Gy, and 8 Gy of radiation at one hour post-IR using ELISA (Figure 2A), which was then validated using Western blot analysis (Figure 2B). We selected the one hour post-irradiation time point to assess γH2AX because, in human microvascular endothelial cells, its levels reach their peak at this specific time point [64]. Our results indicated that 8 Gy significantly increased H2AX phosphorylation compared to the non-irradiated HMEC-1 (*p* < 0.05), whereas there was no difference between 0.25 Gy and 0 Gy (Figure 2A,B). p53 levels also increase rapidly, such as within hours post-irradiation, in response to radiation-induced DNA damage [65]. Therefore, we analyzed the p53 levels in HMEC-1 exposed to doses ranging from 0.1 Gy to 8 Gy at 4 h post-IR using Western blot analysis, to capture the initial response phase. As expected, high doses upregulated the p53 levels, whereas low doses had no impact on its expression (Figure 2C). In summary, our data indicate that only high doses increased rH2AX and p53 in the irradiated HMEC-1.

### 3.3. High Doses Induce Cell Senescence in Irradiated HMEC-1

Research findings demonstrate that exposure to ionizing radiation can cause DNA damage and oxidative stress in endothelial cells, leading to cell senescence [42,66], which increases the risk of developing cardiovascular diseases [17]. Both p21^CIP1^ (p21) and p16^INK4α^ (p16) likely act as important mediators in the cellular response to radiation-induced DNA damage [67]. By inducing cell cycle arrest, they potentially contribute to the establishment of a senescent phenotype in damaged cells [67]. The effects of p21 and p16 by halting cell division might take longer to manifest fully [65]; therefore, we selected 4 days post-irradiation as the time point to assess whether the cells entered a state of prolonged cell cycle arrest or underwent repair. To determine the extent of cell senescence, we evaluated the activity of the lysosomal enzyme SA-β-gal using a plate-based assay (Figure 3A) and assessed the levels of p21 and p16 using Western blot analysis (Figure 3B,C). Our results indicate that radiation at doses equal to or higher than 4 Gy significantly increased SA-β-gal activity (Figure 3A) and p21 expression (Figure 3B) in irradiated HMEC-1 on day 4 post-IR, relative to the non-irradiated cells. However, the expression of p16 was not affected by radiation, regardless of the radiation dose, in HMEC-1 (Figure 3C). 

### 3.4. High Doses Induce Oxidative Stress in Irradiated HMEC-1

Radiation can damage ECs by generating ROS, which react aggressively and cause damage to various cellular components, like DNA, proteins, and lipids [1]. Using DCFDA, a cell-permeant reagent, we quantified hydroxyl, peroxyl, and other ROS species’ activity in HMEC-1 exposed to doses ranging from 0.25 Gy to 8 Gy on day 4 after IR to check the ongoing state of oxidative stress (Figure 4). When compared to the non-irradiated cells, radiation exposure significantly increased the cellular ROS levels by 53% at 2 Gy (*p* = 0.019), 70% at 4 Gy (*p* = 0.002), and 114% at 8 Gy (*p* < 0.0001) (Figure 4B). Meanwhile, 1 Gy elevated the cellular ROS levels by 33% compared to 0 Gy, but the difference was not significant (*p* = 0.19) (Figure 4B). Notably, low doses did not stimulate ROS production in HMEC-1 (Figure 4B). These findings indicate that radiation-induced ROS production in HMEC-1 is dose-dependent at 4 days post-radiation.

### 3.5. High Doses Induce Mitochondrial Dysfunction in Irradiated HMEC-1

Radiation can cause damage to the mitochondria, which play a crucial role in producing energy and generating ROS [41,47]. To investigate this further, we measured the mitochondrial potential (ΔΨm) using a cationic fluorophore, TMRE, in HMEC-1 exposed to doses ranging from 0.25 Gy to 8 Gy. We found that high doses of radiation led to a significant dose-dependent decrease in ΔΨm (Figure 5). Specifically, at doses of 1 Gy, 2 Gy, 4 Gy, and 8 Gy, ΔΨm decreased to 76.1% ± 4.0%, 62.9% ± 1.8%, 51.3% ± 0.5%, and 51.2% ± 2.5% of the control level, respectively (*p* < 0.05, Figure 5B). Notably, low doses of radiation (<1 Gy) did not affect mitochondrial function. These results suggest that mitochondrial dysfunction induced by radiation is dose-dependent.

### 3.6. Radiation at Varying Doses Impacts Mitochondrial Morphology in Irradiated HMEC-1

We previously showed that high doses of radiation decreased the ΔΨm. Changes in mitochondrial morphology can impact mitochondrial function; therefore, we investigated whether radiation at varying doses ranging from 0.25 Gy to 8 Gy would impact the mitochondrial morphology in HMEC-1 cells. To assess the mitochondrial morphology, we used the standard morphology method and categorized their shapes in an individual cell into three groups: elongated, tubular, and fragmented/swollen mitochondria (Figure 6A). In the non-irradiated HMEC-1 cells, the majority (90%) of cells contained tubular mitochondria, while elongated and fragmented/swollen mitochondria were present in only 5% of cells each (Figure 6B–D). Upon exposure to radiation, all doses significantly increased the percentage of elongated mitochondria, with low doses (<1 Gy) having a greater effect than high doses (≥1 Gy) (Figure 6B). However, only high doses caused a notable increase in mitochondrial fragmentation/swelling (Figure 6C). Consequently, the percentage of tubular mitochondria, which made up the majority in the non-irradiated cells, decreased in a dose-dependent manner (Figure 6D).

### 3.7. The Impact of Radiation at Varying Doses on Mitochondrial Fusion and Fission Machinery in Irradiated HMEC-1

The impact of gamma radiation on mitochondrial fusion and fission machinery in irradiated HMEC-1 was investigated by examining the expression levels of mitofusin isoforms (Mfn1 and Mfn2), optic atrophy 1 (Opa1), and dynamin-related protein 1 (Drp1), which are large GTPases from the dynamin family that regulate the overall mitochondrial morphology [68,69,70]. The examination was conducted on day 4 post-IR. The combination of Mfn1, Mfn2, Opa1, and Drp1 determines the mitochondrial morphology [71]. Western blot analysis was used to determine the expression levels of these proteins (Figure 7). Radiation at doses of 0.5 Gy and 1 Gy was found to increase Mfn1 expression (Figure 7A,C), while higher doses (2 Gy and greater) significantly decreased Opa1 expression in a dose-dependent manner (Figure 7A,E). There was also a trend towards a decrease in Opa1 at 1 Gy, although this did not reach statistical significance (*p* = 0.07, Figure 7A,E). The Mfn2 (Figure 7A,D) and Drp1 (Figure 7B,F) expression levels did not differ significantly across all doses tested. Our data suggest that lower-dose-induced mitochondrial elongation may be due to the upregulation of Mfn1 expression, while higher-dose-triggered mitochondrial fragmentation may be caused by the downregulation of Opa1 expression.

## 4. Discussion

It is critical to better understand the damage mechanisms in both high- and low-dose radiation exposure. It is widely acknowledged that mitochondria are highly sensitive to radiation exposure [39,40] and that radiation of high (>1 Gy) doses and different qualities can result in diverse changes in mitochondrial morphology [72,73,74]. However, the exact roles played by the mitochondrial morphology in determining endothelial cell fate after radiation exposure remain unclear. In this study, we demonstrated that exposing HMEC-1 to high doses of radiation resulted in reduced survival, which may be attributed to factors such as DNA damage, oxidative stress, cell senescence, and mitochondrial dysfunction. Our results further indicated that low doses induced a small but significant increase in cell survival, and that this was achieved without detectable DNA damage, oxidative stress, cell senescence, or mitochondrial dysfunction in HMEC-1. More importantly, our study suggests that changes in mitochondrial morphology are likely involved in the mechanism for the radiation dose-dependent effect on the survival of microvascular endothelial cells, thus contributing to a better understanding of the relationship between radiation doses and cellular effects.

In this study, we report mitochondrial fragmentation and swelling at high doses of radiation in HMEC-1, which is consistent with previous studies that utilized normal human fibroblast-like cells [74], human cervical carcinoma cell line HeLa [72], and primary hippocampal neurons prepared from embryonic day 18 in Sprague-Dawley rats [73]. When exposed to high levels of stress, such as radiation at high doses, mitochondria segregate into smaller, separate mitochondrial organelles, referred to as mitochondrial fission, to remove damaged mitochondria through autophagy [71,75]. Radiation-induced (equal to and greater than 1 Gy) mitochondrial fission has been reported to be regulated by the recruitment of Drp1 from the cytosol to the mitochondria through Drp1 phosphorylation at the serine 616 residue, which was assessed in cancer cells at 12–24 h post-IR by Jin et al. [72]. In contrast, our study revealed that high doses induced mitochondrial fission by a decrease in Opa1 assessed on day 4 post-IR. Opa1 is primarily located in the inner mitochondrial membrane, and it plays a role in regulating the fusion of the mitochondrial membrane [76]. A reduction in its levels can trigger mitochondrial fission [77]. However, our study revealed that there were no significant differences in total DRP1 expression levels across varying radiation doses. The discrepancy between our result and the previously published study may be attributed to variations in the timing of assessment and the immortalized versus carcinogenic status of the cells. In addition to total Drp1 expression, it is important to note that Drp1 can undergo activation through post-translational modifications, such as phosphorylation. Upon activation, Drp1 translocates to the outer mitochondrial membrane and assembles into a ring complex, thereby constricting the membrane to initiate fission [75]. Subsequent investigations are required to validate and explore this mechanism further. Notably, previous studies have suggested that only low doses can induce mitochondrial fusion [73], but our data showed that radiation, regardless of the dose, promoted mitochondrial elongation in HMEC-1, with low doses resulting in more elongation than high doses. In response to radiation exposure, mitochondria may fuse together to maintain the integrity of mitochondrial DNA, mitochondrial respiration, and the equilibration of the mitochondrial membrane potential [78], as an adaptive response to cellular stress [78]. We suggest that increased mitochondrial fusion is the underlying mechanism for the improved survival of irradiated HMEC-1 at low doses. However, at high doses, mitochondria underwent not only fusion but also fission, and excessive mitochondrial fission can result in cellular damage due to increased oxidative stress and mitochondrial dysfunction [79]. 

Radiation-induced endothelial senescence is a significant phenomenon with important implications for vascular health and tissue damage associated with radiation exposure [2]. Previous studies have shown that exposure to high doses of radiation (10 Gy) primarily induces endothelial cell senescence or permanent cell cycle arrest via the DNA double-strand break (DSB)–p53–p21 pathway [17]. Our data showed that 8 Gy led to a notable increase in H2AX phosphorylation at one hour post-IR. Furthermore, there was significant upregulation in the expression of p53 and p21 at four hours and four days post-IR, respectively. These findings provide additional evidence of the crucial role of the DSB–p53–p21 pathway in the initiation of senescence in endothelial cells triggered by an 8 Gy dose. 

Wound healing is known to be delayed by high doses of radiation, at least partially due to radiation-induced damage to blood vessels [80,81]. The relationship between low doses of radiation and wound healing, however, is not straightforward. Depending on factors such as the duration of exposure, the severity of the wound, and the individual’s overall health, low doses of radiation could either positively or negatively affect wound healing [21,82,83,84]. The findings from this study in HMEC-1, a group of human dermal microvascular ECs, may encourage further research to investigate the effects of acute exposure to low doses of radiation on angiogenesis in wounds and the healing process.

An interesting finding of this research is that HMEC-1 benefit from low-dose radiation exposure in terms of their survival only under prompt low-dose exposure (less than 1 Gy) conditions. Previous studies using human aortic endothelial cells showed that gamma low-dose rate radiation stimulated adaptive responses in a dose- and dose-rate- dependent manner [85]. Although not the focus of this report, it is essential to investigate the mechanisms of radiation-induced endothelial cell dysfunction to better understand the impact of both dose- and dose-rate-dependent radiation effects on cardiac disease development. Although chronic exposure to low doses of radiation is generally thought to be less harmful than acute exposure to high doses of radiation, it may still increase the risk of health issues over time [86]. Examples of chronic exposure to low doses include cosmic or environmental radiation, repeated medical procedures involving radiation, and occupational exposure for those working in nuclear power plants, medical facilities, and scientific research facilities [86]. As a result, it is crucial to investigate the differences in the effects of radiation on ECs resulting from chronic versus acute exposure to low doses of radiation. 

The HMEC-1 cell line, being a permanently established cell line, has undergone genetic modifications from its original primary cells [87]. We acknowledge the potential impact of these modifications on the cell line’s response to irradiation. Additionally, we recognize the inherent disparities between monolayer-cultured cells and cells in their native environment, including differences in cell–cell interactions [88]. These variations may significantly influence the radiation response and are important considerations when interpreting our findings.

It is worth noting that, in our effort to explore multiple endpoints, including β-gal activity, p21, p16, ROS, mitochondrial membrane potential, and mitochondrial morphology, across varying doses, we conducted a comprehensive time course study including days 1, 4, 7, and 14 post-IR. This extensive exploration aimed to identify the optimal time point for these specific endpoints. Based on our preliminary investigations derived from these pilot studies, we have strong indications that day 4 post-IR stands out as the optimal time point for the evaluation of these specified endpoints.

## 5. Conclusions

Taken together, the results of this study indicate that alterations in mitochondrial morphology play a crucial role in determining the effect of radiation on microvascular endothelial cells. Understanding these responses is crucial in the context of the cardiovascular system because endothelial cell dysfunction is a hallmark of various cardiovascular diseases [51]. Radiation-induced damage to these cells could potentially contribute to cardiovascular complications in individuals exposed to radiation. This research, by delineating the specific mechanisms through which radiation affects endothelial cells, offers invaluable insights into the potential impact of radiation exposure on cardiovascular health. This understanding could lay the groundwork for the development of protective strategies or treatments aimed at mitigating radiation-induced cardiovascular complications. Given the critical involvement of mitochondria in radiation-induced cellular responses, strategies tailored to target mitochondria specifically could hold promise in ameliorating the adverse effects of radiation exposure.

## Figures and Tables

**Figure 1 cells-13-00039-f001:**
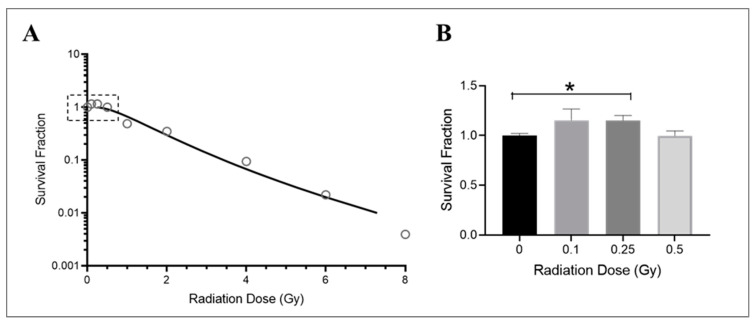
Dose–response effect on the survival of irradiated HMEC-1. (**A**) The dose–survival curve (based on 14-day colony formation data) shows a dose-dependent decrease in the number of colonies at high doses (equal to and greater than 1 Gy). An inset graph in Figure 1A demonstrates the relationship between low doses (less than 1 Gy) and cell survival when the y-axis is set as a logarithmic scale. (**B**) (the y-axis is set as a linear scale): 0.25 Gy increased survival of HMEC-1 compared to that of non-irradiated cells. The study had N = 3–32 per radiation dose, and one-way ANOVA and Dunnett’s multiple comparisons test indicated a statistically significant difference (* *p* < 0.05) for 0.25 Gy vs. 0 Gy. Data in (**B**) are expressed as mean + SEM.

**Figure 2 cells-13-00039-f002:**
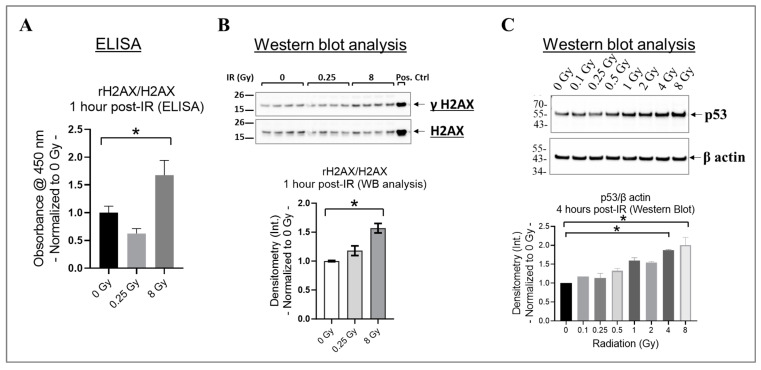
The effect of radiation on H2AX phosphorylation at the serine 139 residue (γH2AX) and p53 expression in irradiated HMEC-1. (**A**) At 1 h post-irradiation (IR), an increase in the ratio of γH2AX to H2AX was observed with 8 Gy radiation, as measured by ELISA. (**B**) This finding was further verified by Western blot analysis, where 8 Gy increased the ratio of γH2AX to H2AX. The protein bands’ intensity was denoted by Int., while the positive control used was the lysate obtained from Jurkat cells exposed to camptothecin (CPT) at 16 h and 37 °C. The study had a sample size of N = 4 for each radiation dose, and statistical significance was calculated by one-way ANOVA and Dunnett’s multiple comparisons test, with * *p* < 0.05 indicating significance between 8 Gy and 0 Gy radiation. (**C**) High doses upregulated p53 levels at 4 h post-IR. The study had a sample size of N = 2 for each radiation dose, with statistical significance calculated by the Kruskal–Wallis test followed by Dunn’s multiple comparisons test, with * *p* < 0.05 for 4 Gy/8 Gy vs. 0 Gy. The data presented here were normalized to 0 Gy radiation. Data are expressed as mean + SEM.

**Figure 3 cells-13-00039-f003:**
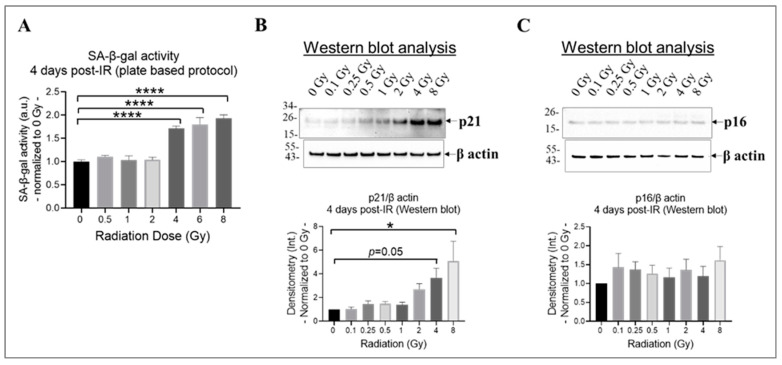
The impact of doses on cell senescence in irradiated HMEC-1 on day 4 post-IR. (**A**) Higher doses of radiation (equal to and greater than 4 Gy) increased SA-β-gal activity, which was measured using a plate-based protocol. SA-β-gal refers to senescence-associated beta-galactosidase. The number of samples used for each radiation dose was 3 to 6. The statistical analysis revealed a significant difference (**** *p* < 0.0001) between the higher doses (4 Gy, 6 Gy, and 8 Gy) and the control (0 Gy), as determined by one-way ANOVA and Dunnett’s multiple comparisons test. (**B**) Higher doses of radiation upregulated the expression of p21, as assessed by Western blot analysis. The number of samples used for each radiation dose was 6. The statistical analysis revealed a significant difference (* *p* < 0.05) between 8 Gy and 0 Gy, as well as a marginal difference (*p* = 0.05) between 4 Gy and 0 Gy, as determined by one-way ANOVA and Dunnett’s multiple comparisons test. (**C**) Radiation, regardless of the dose, did not affect the expression of p16, as assessed by Western blot analysis. The number of samples used for each radiation dose was 6. In Western blot analyses (**B**,**C**), the loading control was β actin and the intensity of protein bands was denoted as Int. The data shown in all three panels were normalized to 0 Gy. Data are expressed as mean + SEM.

**Figure 4 cells-13-00039-f004:**
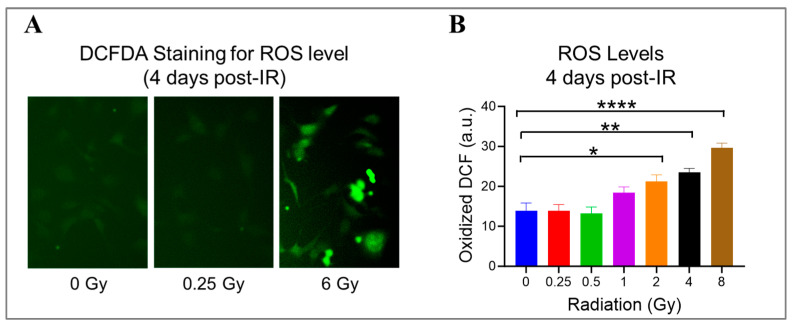
The impact of doses on cellular levels of reactive oxygen species (ROS) in irradiated HMEC-1 cells. (**A**) Representative images of cellular ROS levels (indicated by green fluorescence) are shown, which were measured using dichlorodihydrofluorescein diacetate (DCFDA) on day 4 after IR. (**B**) The quantification of ROS levels reveals that high radiation doses led to an increase in cellular ROS levels in a dose-dependent manner. The measurement units used were arbitrary units (a.u.), and the experiment was conducted with 3 samples for each radiation dose. Statistical analysis using one-way ANOVA and Dunnett’s multiple comparisons test showed significant differences between radiation doses, as indicated by asterisks. Specifically, * *p* < 0.05 for 2 Gy vs. 0 Gy, ** *p* < 0.01 for 4 Gy vs. 0 Gy, and **** *p* < 0.0001 for 8 Gy vs. 0 Gy. Data in (**B**) are expressed as mean + SEM.

**Figure 5 cells-13-00039-f005:**
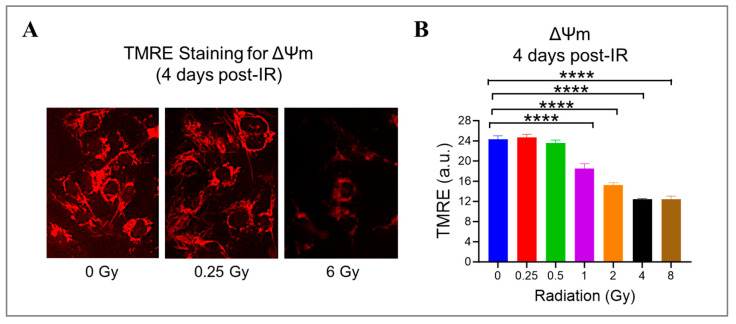
The effect of gamma radiation doses on mitochondrial membrane potential (ΔΨm) in irradiated HMEC-1. (**A**) Representative images of ΔΨm (red fluorescence) assessed by tetramethylrhodamine ethyl ester (TMRE) on day 4 post-IR are shown. (**B**) The quantification of ΔΨm indicated that there was a radiation dose-dependent decline in ΔΨm at high doses. The results were reported in arbitrary units (a.u.), and there were 3 samples per radiation dose. Using one-way ANOVA and Dunnett’s multiple comparisons test, the statistical analysis found that there was a significant difference (**** *p* < 0.0001) between 1 Gy/2 Gy/4 Gy/8 Gy and 0 Gy. Data are expressed as mean + SEM.

**Figure 6 cells-13-00039-f006:**
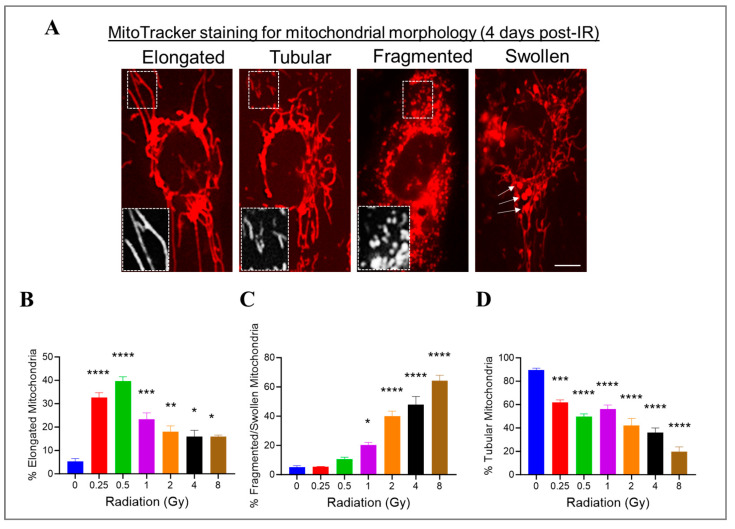
The effects of doses on mitochondrial morphology in irradiated HMEC-1. (**A**) Representative images of three typical mitochondrial shapes, elongated, tubular, and fragmented/swollen, are shown, which were detected using MitoTracker Red chloromethyl-X-rosamine (CMXRos) on day 4 post-IR. (**B**) Radiation at doses ranging from 0.25 Gy to 8 Gy promoted mitochondrial elongation. (**C**) High doses triggered mitochondrial fragmentation/swelling. (**D**) Radiation ranging from 0.25 Gy to 8 Gy resulted in a decrease in tubular mitochondria. There were 3 samples per radiation dose, and the statistical analysis conducted using one-way ANOVA and Dunnett’s multiple comparisons test revealed significant differences between irradiated cells and control cells, denoted by * *p* < 0.05; ** *p* < 0.01; *** *p* < 0.001; and **** *p* < 0.0001. Data are expressed as mean + SEM.

**Figure 7 cells-13-00039-f007:**
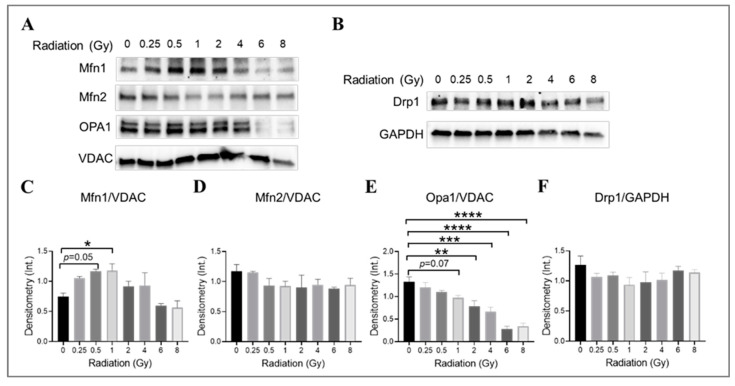
The impact of doses on mitochondrial fusion and fission machinery in irradiated HMEC-1. The examination was conducted on day 4 post-IR. (**A**,**C**) 1 Gy and 0.5 Gy upregulated Mfn1 expression, as determined by Western blot analysis. However, no differences in Mfn2 expression were observed across all doses tested, as shown in panels (**A**,**D**). (**A**,**E**) High doses decreased Opa1 expression. (**B**,**F**) There were no differences in Drp1 expression observed across all doses tested. There were three samples per radiation dose, and the statistical analysis conducted using one-way ANOVA and Dunnett’s multiple comparisons test revealed significant differences between irradiated cells and control cells, denoted by * *p* < 0.05; ** *p* < 0.01; *** *p* < 0.001; and **** *p* < 0.0001. The mitochondrial loading control in Western blot analyses was the voltage-dependent anion channel (VDAC), whereas the cytosol loading control was glyceraldehyde-3-phosphate dehydrogenase (GAPDH). The abbreviations used in the figures are as follows: Mfn1 represents mitofusin 1; Mfn2 represents mitofusin 2; Opa1 represents optic atrophy 1; and Drp1 represents dynamin-related protein 1. Data are expressed as mean + SEM.

## Data Availability

The data presented in this study are available on request from the corresponding author.

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
