# Peer review of "Dose-Dependent Effects of Radiation on Mitochondrial Morphology and Clonogenic Cell Survival in Human Microvascular Endothelial Cells"

_cells, 2023, doi:10.3390/cells13010039_

Round 1
Reviewer 1 Report
Comments and Suggestions for Authors
This is a well written, well thought out study, however there are several issues that need to be clarified/addressed.
1) there needs to be a justification for the post-IR treatment times of analysis. these authors choose 1 hr and 4 days post treatment while work in the literature cited used different times. did they use a time study to establish maximal modifications?
2) the data in Figures 2, 3, and 5 represent only 2-3 data points. the results indicate a trend but cannot fairly be analyzed statistically. ANOVA analysis presumes that the data for each group yields a mean of random samples for that group. two data points cannot meet that objective. the data represents trends but the level of significance cannot be fairly stated. also, there is no indication if the deviation bars in the figures are Standard Deviations or SEMs (standard error of the means). this can make the presentation of the data very different.
3) in the Discussion, line 465 it states "changes in mitochondrial morphology are DIRECTLY involved....on the survival of microvascular ECs." there is no direct evidence, just correlative evidence that IMPLIES that this may be the case.
4) HMEC-1 cell line is a permanently established cell line and has therefore undergone some genetic modifications from the original primary cells. the Discussion should address that genetic modifications may alter responses to IR. also, cultured cells in monolayers are very different from cells in-situ where cell contacts with homogeneous and heterogeneous cells might mitigate or enhance responses to IR treatments.
Author Response
We are grateful for the Reviewer's commendations regarding the significance, design, and presentation of our study, along with the insightful suggestions that have contributed to enhancing our manuscript. Below please find point-by-point responses addressing the Reviewer's. All revisions made to the manuscript are highlighted in yellow.
Reviewer 1:
This is a well written, well thought out study, however there are several issues that need to be clarified/addressed.
Comment 1: there needs to be a justification for the post-IR treatment times of analysis. these authors choose 1 hr and 4 days post treatment while work in the literature cited used different times. did they use a time study to establish maximal modifications?
Response 1: We agree with this comment. In Figure 2A and 2B, our data demonstrate an 8 Gy-induced increase in γH2AX levels at 1-hour post-IR in HMEC-1 cells. Initially, we referenced a study (DOI: 10.4103/genint.genint_1_22) investigating γH2AX levels in human blood samples, observing changes 0.5 hours after radiotherapy in breast cancer patients. However, we recognize the relevance of another study (DOI: 10.1667/RR0975.1) conducted by Kataoka et al., which is more aligned with the context presented in our manuscript. Kataoka et al. identified the most robust γ-H2AX response in human microvascular endothelial cells (HMEC) occurring 1-hour post-irradiation, with diminishing frequencies observed at 4 and 24 hours later. Thus, in our revised manuscript, we have appropriately cited this study (lines 311-312) as it closely aligns with our research focus.
In our effort to explore multiple endpoints, including ROS (Figure 3), β-gal activity, p21, p16 (Figure 4), mitochondrial membrane potential (Figure 5), and mitochondrial morphology (Figure 6 and 7), across varying doses, we conducted a comprehensive time course study including days 1, 4, 7, and 14 post-IR. This extensive exploration aimed to identify the most effective time point for maximal modifications across these specific endpoints. Based on our preliminary investigations derived from these pilot studies, we have strong indications that day 4 post-IR stands out as the optimal time point for evaluating these specified endpoints.
Comment 2: the data in Figures 2, 3, and 5 represent only 2-3 data points. the results indicate a trend but cannot fairly be analyzed statistically. ANOVA analysis presumes that the data for each group yields a mean of random samples for that group. two data points cannot meet that objective. the data represents trends but the level of significance cannot be fairly stated. also, there is no indication if the deviation bars in the figures are Standard Deviations or SEMs (standard error of the means). this can make the presentation of the data very different.
Response 2: Thank you for the feedback on Figures 2, 3, and 5. We've considered your observations regarding the statistical analysis and data presentation.
Your perspective on conducting ANOVA with only two data points per group is well-founded. We've explored alternative methods for analyzing such data, specifically non-parametric tests like the Kruskal-Wallis test for comparing multiple groups. We have 2 data points for all doses in Figure 2C, therefore, we have conducted a reanalysis of the data presented in Figure 2C using the Kruskal-Wallis test followed by Dunn’s multiple comparisons test. The revised Figures 2C in the manuscript now incorporate these updated findings (lines 188-189, 276-277, and 335).
Regarding Figures 3, 5, and 6, we initially had 2 data points for 0.1 Gy and 3 data points for other doses. Consequently, we have excluded the 0.1 Gy data from the revised versions of Figures 3B, 5B, and 6B-6D (lines 29-31, 189, 242-243, 257, 269, 342, 356-357, 394-395, 406-407, 413, 419-420, and 432).
Additionally, your observation regarding the error bars has been acknowledged. To provide clarity, we've specified in the figure legends that the bars represent Standard Errors of the Means (SEM) (lines 307, 337, 360-361, 390, 409, 435-436,465).
Comment 3: in the Discussion, line 465 it states "changes in mitochondrial morphology are DIRECTLY involved.... on the survival of microvascular ECs." there is no direct evidence, just correlative evidence that IMPLIES that this may be the case.
Response 3: We agree that our findings suggest a correlation rather than direct causation between changes in mitochondrial morphology and the survival of microvascular ECs. The language used in that statement might have overemphasized the direct involvement without explicit direct evidence. We adjusted the wording in the Discussion section to “our study suggests that changes in mitochondrial morphology are likely involved in the mechanism for the radiation dose-dependent effect on the survival of microvascular endothelial cells” (lines 35-37, 477-480), which more accurately reflect this correlation-based implication without implying direct causation.
Comment 4: HMEC-1 cell line is a permanently established cell line and has therefore undergone some genetic modifications from the original primary cells. the Discussion should address that genetic modifications may alter responses to IR. also, cultured cells in monolayers are very different from cells in-situ where cell contacts with homogeneous and heterogeneous cells might mitigate or enhance responses to IR treatments.
Response 4: We agree with the comment. In the revised Discussion section, we added the statement “The HMEC-1 cell line, being a permanently established cell line, has undergone genetic modifications from its original primary cells. We acknowledge the potential impact of these modifications on the cell line's response to irradiation. Additionally, we recognize the inherent disparities between monolayer-cultured cells and cells in their native environment, including differences in cell-cell interactions. These variations may significantly influence the radiation response and are important considerations when interpreting our findings” (lines 548-554).
Reviewer 2 Report
Comments and Suggestions for Authors The topic of the research described in the MS is very relevant for the understanding radiation-induced organ disfunction. In fact, the effects of radiation on endothelial cells in all tissues and organs are of utmost importance. Meanwhile several concerns need to be clarified. Measurement and analysis of indicators characterizing the state of the ECs were carried out at different times after irradiation. For example, b-Gal – at 4 day after IR, H2AX – at 1 h after IR, p53 – at 4 h after IR, p21 – at 4 d after IR, and so on. It is necessary to explain the choice of temporary exposures after irradiation for the measured parameters. In section 3.7 and Fig. 7, the time after irradiation is not indicated at all.Author Response
Thank you for your valuable feedback on the varying time points used for measuring the indicators after irradiation. The selection of times after irradiation for assessing specific markers for characterizing the state of the ECs was mainly based on the biological kinetics of the markers themselves. To enhance the clarity, we have justified the chosen time points in the Result and Discussion sections of the revised manuscript. Also, to improve the flow and coherence, we've re-sequenced Figures 3 and 4.
We added:
(lines 315-317): We selected the one-hour post-irradiation for assessing γH2AX because, in human microvascular endothelial cells, its levels reach their peak at this specific time point (doi:10.1667/RR0975.1).
(lines 319-322): P53 levels also increase rapidly, such as within hours post-irradiation, in response to radiation-induced DNA damage (doi:10.1016/s1367-5931(99)80014-3). Therefore, we analyzed p53 levels in HMEC-1 exposed to doses ranging from 0.1 Gy to 8 Gy at 4 hours post-IR using Western blot analysis to capture the initial response phase.
(lines 344-350): Both p21CIP1 (p21) and p16INK4α (p16) likely act as important mediators in the cellular response to radiation-induced DNA damage (doi:10.3389/fphar.2018.00522). By inducing cell cycle arrest, they potentially contribute to the establishment of a senescent phenotype in damaged cells (doi:10.3389/fphar.2018.00522). The effects of p21 and p16 on halting cell division might take longer to manifest fully (doi:10.1016/s1367-5931(99)80014-3), therefore, we selected 4 days post-irradiation as the time point for assessing whether the cells have entered a state of prolonged cell cycle arrest or have undergone repair.
(lines 565-571): It is worth noting that, in our effort to explore multiple endpoints, including β-gal ac-tivity, p21, p16, ROS, mitochondrial membrane potential, and mitochondrial morphology, across varying doses, we conducted a comprehensive time course study including days 1, 4, 7, and 14 post-IR. This extensive exploration aimed to identify the optimal time point for these specific endpoints. Based on our preliminary investigations derived from these pilot studies (Data not shown), we have strong indications that day 4 post-IR stands out as the optimal time point for evaluating these specified endpoints.
Again, we appreciate your observation regarding the absence of time indication after irradiation in section 3.7 and Fig. 7. We apologize for this oversight. In the revised manuscript, we have added the statement “The examination was conducted on day 4 post-IR” (lines 451, 465) for better clarity.
Reviewer 3 Report
Comments and Suggestions for Authors
In this manuscript, the authors investigate potential mechanisms for the phenomena of low dose radioprotection. The authors identify associations with MFN1 and OPA1 dysregulation and changes in mitochondrial fusion and fission that demonstrate specific changes at low doses vs high doses of radiation. This publication contributes new and interesting insights into this poorly understood area. The manuscript and figures are well written and assembled. Comments that should be addressed specifically are noted below:
Primary Comments:
1 1) The authors should examine DRP-1 expression in greater depth in line with reported . The authors state that DRP-1 is reported as a regulator of mitochondrial fission and that this occurs via DRP-1 phosphorylation which affects localisation (cytoplasm to mitochondria) (lines 475-478). The authors should either examine DRP-1 movement and mitochondrial co-localisation via immunocytochemistry or examine levels of phosphorylated DRP-1 by western analysis to confirm a lack of involvement of this protein/pathway in the radiation effects observed. While OPA1, and MFN1 clearly regulated, the authors have not proven DRP1 does not have a role based just on demonstration of total DRP1 expression levels.
22) With regard to the discussion statement (lines 464-468) - “changes in mitochondrial morphology are directly involved in mechanism for radiation dose-dependent effect on survival of MVEC”. This study has shown an association between survival and mitochondrial alternations (which may be cause or effect) but have not proven causality (there are no knockdown, overexpression studies), please modify language to recognise that direct causality has not been shown here. Similarly moderate the language with regard to similar claims/conclusions in lines 528-530, and also in the abstract lines 35-37. The statement at lines 490-492 “we suggest that increased mitochondrial fusion is the underlying mechanism for improved survival of irradiated microvessels at low doses” is more appropriately worded to suggest the association rather than proven mechanism.
Author Response
We are grateful for the Reviewer's commendations regarding the significance, design, and presentation of our study, along with the insightful suggestions that have contributed to enhancing our manuscript. Below please find point-by-point responses addressing the Reviewer's points. All revisions made to the manuscript are highlighted in yellow.
Reviewer 2:
In this manuscript, the authors investigate potential mechanisms for the phenomena of low dose radioprotection. The authors identify associations with MFN1 and OPA1 dysregulation and changes in mitochondrial fusion and fission that demonstrate specific changes at low doses vs high doses of radiation. This publication contributes new and interesting insights into this poorly understood area. The manuscript and figures are well written and assembled. Comments that should be addressed specifically are noted below:
Primary Comments:
Comment 1: The authors should examine DRP-1 expression in greater depth in line with reported. The authors state that DRP-1 is reported as a regulator of mitochondrial fission and that this occurs via DRP-1 phosphorylation which affects localisation (cytoplasm to mitochondria) (lines 475-478). The authors should either examine DRP-1 movement and mitochondrial co-localisation via immunocytochemistry or examine levels of phosphorylated DRP-1 by western analysis to confirm a lack of involvement of this protein/pathway in the radiation effects observed. While OPA1, and MFN1 clearly regulated, the authors have not proven DRP1 does not have a role based just on demonstration of total DRP1 expression levels.
Response 1: We agree with the reviewer’s comment about Drp1. We examined only total Drp1 expression in this study because of the initial funding scope. We acknowledge this limitation and have included the following discussion in the revised manuscript: “However, our study revealed that there aren't any significant differences in total DRP1 expression levels across varying radiation doses. The discrepancy between our result and previous published study may be attributed to variation in the timing of assessment and the immortalized versus carcinogenic status of the cells. In addition to total Drp1 expression, it is important to note that Drp1 can undergo activation through post-translational modifications, such as phosphorylation. Upon activation, Drp1 translocates to the outer mitochondrial membrane and assembles into a ring complex, thereby constricting the membrane to initiate fission (DOI: 10.3390/antiox12061163). Subsequent investigations are required to validate and explore this mechanism further” (lines 495-504).
Comment 2: With regard to the discussion statement (lines 464-468) - “changes in mitochondrial morphology are directly involved in mechanism for radiation dose-dependent effect on survival of MVEC”. This study has shown an association between survival and mitochondrial alternations (which may be cause or effect) but have not proven causality (there are no knockdown, overexpression studies), please modify language to recognise that direct causality has not been shown here. Similarly moderate the language with regard to similar claims/conclusions in lines 528-530, and also in the abstract lines 35-37. The statement at lines 490-492 “we suggest that increased mitochondrial fusion is the underlying mechanism for improved survival of irradiated microvessels at low doses” is more appropriately worded to suggest the association rather than proven mechanism.
Response 2: We agree that our findings suggest a correlation rather than direct causation between changes in mitochondrial morphology and the survival of microvascular ECs. The language used in that statement might have overemphasized the direct involvement without explicit direct evidence. We adjusted the wording in the Discussion section to “our study suggests that changes in mitochondrial morphology are likely involved in the mechanism for the radiation dose-dependent effect on the survival of microvascular endothelial cells” (lines 35-37, 477-480), which more accurately reflect this correlation-based implication without implying direct causation.
Round 2
Reviewer 1 Report
Comments and Suggestions for Authors
one minor editorial correction: line 35: "...the study suggestses...." should be "suggests"
Comments on the Quality of English Languageone minor correction as noted to authors above
Author Response
Thank you for bringing that editorial oversight to our attention. We have corrected the error on line 35 from 'suggestses' to 'suggests' in the revised version of the manuscript.
Reviewer 3 Report
Comments and Suggestions for Authors
All revisions satisfactory.
Author Response
Thank you for reviewing the revisions. Your feedback has been invaluable in refining the work.